# Autophagy Inhibition in BRAF-Driven Cancers

**DOI:** 10.3390/cancers13143498

**Published:** 2021-07-13

**Authors:** Mona Foth, Martin McMahon

**Affiliations:** 1Huntsman Cancer Institute, Salt Lake City, UT 84112, USA; mona.foth@hci.utah.edu; 2Department of Dermatology, University of Utah, Salt Lake City, UT 84112, USA

**Keywords:** BRAF, MEK1/2, autophagy, targeted therapy, drug resistance, metabolism

## Abstract

**Simple Summary:**

BRAF is a protein kinase that is frequently mutationally activated in cancer. Mutant BRAF can be pharmacologically inhibited, which in combination with blockade of its direct effector, MEK1/2, is an FDA-approved therapeutic strategy for several BRAF-mutated cancer patients, such as melanoma, non-small-cell lung carcinoma, and thyroid cancer. However, therapy resistance is a major clinical challenge, highlighting the need for comprehensive investigations on the biological causes of such resistance, as well as to develop novel therapeutic strategies to improve patient survival. Autophagy is a cellular recycling process, which has been shown to allow cancer cells to escape from BRAF inhibition. Combined blockade of autophagy and BRAF signaling is a novel therapeutic strategy that is currently being tested in clinical trials. This review describes the relationship between BRAF-targeted therapy and autophagy regulation and discusses possible future treatment strategies.

**Abstract:**

Several BRAF-driven cancers, including advanced BRAF^V600E/K^-driven melanoma, non-small-cell lung carcinoma, and thyroid cancer, are currently treated using first-line inhibitor combinations of BRAF^V600E^ plus MEK1/2. However, despite the success of this vertical inhibition strategy, the durability of patient response is often limited by the phenomenon of primary or acquired drug resistance. It has recently been shown that autophagy, a conserved cellular recycling process, is increased in BRAF-driven melanoma upon inhibition of BRAF^V600E^ signaling. Autophagy is believed to promote tumor progression of established tumors and also to protect cancer cells from the cytotoxic effects of chemotherapy. To this end, BRAF inhibitor (BRAFi)-resistant cells often display increased autophagy compared to responsive lines. Several mechanisms have been proposed for BRAFi-induced autophagy, such as activation of the endoplasmic reticulum (ER) stress gatekeeper GRP78, AMP-activated protein kinase, and transcriptional regulation of the autophagy regulating transcription factors TFEB and TFE3 via ERK1/2 or mTOR inhibition. This review describes the relationship between BRAF-targeted therapy and autophagy regulation, and discusses possible future treatment strategies of combined inhibition of oncogenic signaling plus autophagy for BRAF-driven cancers.

## 1. The BRAF→MEK1/2→ERK1/2 MAP Kinase Signaling Pathway and FDA-Approved Pathway-Targeted Therapy

BRAF encodes a serine-specific protein kinase that is a key transducer of cellular signaling pathways that influence cell proliferation, differentiation, and mature cell function [1]. However, *BRAF* is also frequently mutated in numerous different cancer types, including ~100% of hairy cell leukemia, 50% of melanoma, 20–40% of thyroid carcinoma, 10–15% of colorectal adenocarcinoma, 10% of glioblastoma, 4% of non-small-cell lung cancer, and small percentages of non-Hodgkin’s lymphoma, multiple myeloma, and ovarian cancer [2,3,4,5,6,7].

Activating point mutations or small in-frame deletions observed in *BRAF*-mutated cancers generally occur within sequences that encode the protein kinase domain [2]. The most frequent *BRAF* mutation (T1799A) encodes BRAF^V600E^, in which the substitution of valine to glutamic acid at amino acid 600 within the activation loop renders BRAF kinase activity constitutively active, with up to 500-fold increased activity [2,8]. Although less common than activating point mutations, there have also been BRAF^V600E^ amplification, truncation events, or splice variants described, as well as a number of *BRAF* gene fusion events [9,10,11].

Mutationally activated BRAF^V600E^ drives the activation of the downstream MEK1/2→ERK1/2 mitogen-activated protein kinase (MAPK) signaling pathway. BRAF directly phosphorylates to activate the dual specificity protein kinases MEK1 and MEK2 on a pair of neighboring serines (S218 and S222 in MEK1 and S222 and S226 in MEK2). In turn, activated MEK1/2 phosphorylate to activate both ERK1 and ERK2 on neighboring threonine and tyrosine residues (T202 and Y204 in ERK1 and T185 and Y187 in ERK2) (Figure 1) [2,12]. Constitutive activation of the BRAF→MEK→ERK pathway in cancer cells provides signals that promote the cell division cycle, suppress programmed cell death/apoptosis, and contribute to numerous other hallmarks of cancer [13,14,15,16,17].

BRAF^V600E^ oncoprotein kinase signaling has been subject to pharmacological targeting in a number of cancers. For example, in melanoma, there are currently three FDA-approved combinations of inhibitors of BRAF^V600E^ plus MEK1/2: dabrafenib plus trametinib (Novartis), vemurafenib plus cobimetinib (Genentech/Plexxicon/Roche), and encorafenib plus binimetinib (Pfizer), based on high response rates of 50–80% (Figure 1) [18,19,20,21,22]. BRAF^V600E^-driven lung and thyroid cancers are also responsive to combined inhibition of BRAF^V600E^ plus MEK1/2 using dabrafenib and trametinib, which are now FDA-approved standard of care for metastatic NCSLC and thyroid cancer patients [23,24]. BRAF^V600E^-driven gliomas displayed variable sensitivity to vemurafenib depending on the histological subtype [25]. In BRAF^V600E^-driven hairy cell leukemia, vemurafenib has shown promising results in patients who were treated off-label [26], but no BRAF^V600E^ inhibitor has been FDA-approved for this hematological malignancy. In stark contrast, results from a phase II clinical trial indicated that BRAF^V600E^-driven colorectal cancer (CRC) is largely refractory to BRAF^V600E^ inhibitors [27]. Common mechanisms of intrinsic or acquired resistance of colorectal cancer to BRAF- or MEK1/2-inhibitors involve re-activation of the MAPK pathway, e.g., through EGFR-mediated activation of RAS and CRAF, or amplification of the *BRAF* gene [28,29,30,31], although other mechanisms of resistance involving alterations of cancer cell metabolism can also play an important role.

Despite the success of BRAF^V600E^ plus MEK1/2-targeted therapy in melanoma, metastatic NCSLC, and thyroid cancer, the durability of patient response is limited either by primary or acquired chemoresistance [32,33]. Several resistance mechanisms have been described, which often involve re-activation of the MEK1/2→ERK1/2 MAPK pathway, e.g., through changes to the BRAF protein itself, such as dimerization of aberrantly spliced BRAF^V600E^ [9,10,11,34], BRAF amplification [35], BRAF fusions [10,11], BRAF kinase domain duplications [36], or activation of the RAF family member CRAF [37,38]. Moreover, point mutations in RAS or MEK1 [34,39,40,41,42,43], as well as upregulation of EGFR, PDGFR, or FGFR3 [44,45], or persistent formation of the eIF4F translation initiation complex [46], have been described to result in re-activation of the MAPK pathway and therapy resistance. Furthermore, PI3K→AKT pathway alterations have been reported to cause resistance to BRAF^V600E^ plus MEK1/2 inhibition, including point mutations in AKT1 and 3 [47,48,49], or loss of the tumor suppressor PTEN [50]. Other reported resistance mechanisms include loss of STAG2 and STAG3 [51] or miRNA upregulation [52]. Therapeutic strategies to forestall or overcome drug resistance are currently being heavily investigated, ranging from discontinuous (“drug holiday”) dosing to combinatorial therapy using inhibitors that induce synergistic killing of tumor cells or otherwise prevent re-activation of the MAPK pathway [53,54].

## 2. BRAF^V600E^-Mediated Metabolic Rewiring Undermines the Efficacy of Pathway-Targeted Therapies

Metabolic reprogramming is one of the emerging hallmarks of cancer and a common feature of many cancer types [17]. Recent studies suggest that oncogenic signaling induces reprogramming of several processes in cellular metabolism, resulting in measurable changes in the flux of various biochemical pathways, including autophagy, glycolysis, glutamine metabolism, mitochondrial respiration, reactive oxygen species (ROS) production, and protein biosynthesis [55,56,57,58,59,60]. Oncogenic BRAF and its immediate upstream activator RAS are reasonably well-described orchestrators of such metabolic transformation, which allows tumors to escape from signaling pathway-targeted therapies (Figure 1) [60,61,62,63,64,65,66,67].

One of such metabolic transformations involves glycolysis. *BRAF*- or *KRAS*-driven colorectal cancer and *BRAF*-driven melanoma exhibit increased glycolysis [58,68], and inhibition of BRAF^V600E^ suppressed the levels of glycolysis in BRAF-driven melanoma [69]. Furthermore, studies have shown that oncogenic BRAF signaling negatively regulates oxidative metabolism towards aerobic glycolysis in melanoma, and BRAF-targeted therapy subsequently increased levels of oxidative phosphorylation (OXPHOS) [59], a metabolic adjustment that can potentially be exploited for therapeutic intervention. A recent study showed that melanoma subsets with elevated levels of OXPHOS were resistant to the MEK1/2 inhibitor selumetinib, and these cells could be sensitized to MEKi with the addition of an mTORC1/2 inhibitor [70]. Moreover, mitochondrial ROS are generated during OXPHOS at the inner mitochondrial membrane, and are indicative of the mitochondrial function [71]. High levels of mitochondrial ROS that are produced during OXPHOS can activate apoptosis and autophagy pathways [72]. Importantly, ROS levels were increased upon MAPK pathway blockade in BRAF-driven melanoma cells using a BRAF inhibitor [73], which may be exploited as a strategy to induce caspase-independent apoptosis programs [74,75,76]. Interestingly, our lab recently showed that NRAS/BRAF-driven melanoma, pancreatic, or colorectal cancer cell lines increase a metabolic process, called autophagy, upon MEK1/2 inhibition, a process we will discuss in the following section [77].

Collectively, these studies indicate that BRAF^V600E^-driven metabolic reprogramming can undermine the efficacy of MAPK pathway-targeted therapy. Hence, inhibition of certain metabolic pathways may sensitize cancer cells to pathway-targeted therapy.

## 3. Autophagy Is a Recycling Mechanism with Tumor-Suppressive or Tumor-Promoting Roles in a Context-Dependent Manner

Macroautophagy (hereafter autophagy) is a metabolic process of cellular self-digestion in which cellular components, such as abnormal protein aggregates and damaged organelles, serve as recycling substrates in order to generate energy or metabolic precursors, often in response to external stress (Figure 2) [78]. Indeed, autophagy is activated upon nutrient deprivation, pathogen infection, hypoxia, as well as in cancer cells as a pro-survival mechanism [78]. Autophagy has a tumor-promoting role in established tumors, however, there is evidence of tumor-suppressive functions in a context-dependent manner [79,80,81,82]. For example, initially during early tumorigenesis, BRAF-driven lung tumor formation was accelerated upon *Atg7* deletion, but later, these tumors slowed their growth and failed to transition from benign to malignant tumors [83].

The protein kinase complex mTORC1 is a critical mediator of alterations in nutrients or mitogenic signals and serves to integrate such signals into appropriate cell biological processes. In that regard, mTORC1 signaling is a key regulator of protein synthesis, autophagy, glucose homeostasis, and lipid and nucleotide synthesis (Figure 2). Two of the best characterized roles of mTORC1 are as part of: (1) the cellular response to alterations in key amino acids, and (2) the response of cells to polypeptide growth factors acting through transmembrane receptor tyrosine kinases [84]. In the former situation, mTORC1 activity is regulated by the Ragulator complex, which serves as a sensor of lysosomal arginine. The Ragulator consists of five LAMTOR (1–5) subunits and the low molecular weight RAS family GTPases RAG-A/RAG-B and RAG-C/RAG-D that form heterodimers [85,86]. The Ragulator complex associates with lysosomal membranes, where amino acid stimulation results in GTP-exchange of the RAG heterodimers [87,88,89]. This allows the Ragulator complex to recruit the mTORC1 complex to the lysosomal membrane via the mTORC1 subunit RAPTOR [89]. mTORC1 can also be activated through cytosolic amino acids, e.g., leucine and arginine, which release repressors such as CASTOR1 (Cellular Arginine Sensor for mTORC1) from the GATOR1 and GATOR2 complexes [89]. In the context of growth factor stimulation, activation of receptor tyrosine kinase signaling leads to activation of the RAS→RAF→MEK→ERK MAP kinase and the PI3′-lipid kinase (PI3K)→AKT protein kinase signaling pathways. AKT is reported to phosphorylate and thereby inactivate the TSC1/2 complex, which acts as a GAP for the RAS family GTPase RHEB, leading to accumulation of RHEB-GTP. RHEB-GTP is reported to bind and promote activation of mTORC1, which then binds to the Ragulator-RAG complex on lysosomal membranes, where it remains active under conditions of non-limiting amino acids [84]. In addition, AKT phosphorylates to inactivate PRAS40, which contributes to mTORC1 activation [90]. Regulation of mTORC1 activity by the RAF→MEK→ERK pathway is reported to be through direct ERK2-mediated phosphorylation of TSC2 [91], which results in dissociation of the TSC1-TSC2 complex, leading to accumulation of RHEB-GTP and thereby increased mTORC1 protein kinase activity. Hence, such observations may explain the coordinate control of mTORC1 protein kinase activity by two parallel signaling pathways that are well-credentialed as bona fide oncogenic signaling pathways in many types of cancer [92,93,94].

An abundance of data links mTORC1 signaling to the regulation of autophagy. For example, when certain key nutrients such as leucine are in abundance, mTORC1′s kinase leads to direct phosphorylation of a master regulator of autophagy, the serine/threonine protein kinase ULK1/ATG1, on serine 757 (pS757) (Figure 2). This phosphorylation disrupts the interaction between ULK1 and the AMP-activated protein kinase (AMPK), and thereby represses autophagy [95]. However, in times of leucine deprivation, mTORC1 activity is decreased, initially leading to decreased pS757-ULK1 phosphorylation that, in turn, leads to increased AMPK→ULK1 signaling [78]. Activated ULK1 forms a complex with ATG13, ATG101, and FIP200, which activates a class-III PI3K complex comprising VPS34, Beclin1, p150, and ATG14, which in turn initiates phagophore membrane elongation towards autophagosome formation (Figure 2) [78]. Indeed, ULK1 is reported to phosphorylate many of these proteins, including ATG13, FIP200, Beclin1, and VPS34, as part of its master regulation of autophagy initiation. The final formation of autophagosomes is promoted by the conjugated ATG gene family members ATG-5, -12, as well as by the ATG8/LC3-II ubiquitin-like conjugation pathways, which require ATG-7 and -10 [78]. Cytosolic LC3 (LC3-I) is cleaved by the ATG4B cysteine protease to reveal glycine 120 as the new C-terminus, which is then conjugated to phosphatidylethanolamine (PE) through the activity of ATG7/3, forming an LC3-PE complex (LC3-II) (Figure 2) [96]. LC3-II is required for autophagic cargo recognition and fusion of the autophagosome with the lysosome. Through ATG5/12 activity, LC3-II associates with both the inner and outer membranes of the autophagosome [97]. Then, upon fusion of the autophagosome with the lysosome, LC3-II is degraded by lysosomal proteases. Consequently, the ratio of cytosolic LC3-I to membranous LC3-II may be used as a biomarker of changes in autophagic activity [98]. However, interpretation of the ratio of LC3-I to LC3-II as a measure of autophagy can be challenging since high autophagic flux results in rapid conversion of LC3-I to LC3-II by proteolysis followed by PE lipidation, as described above. Inhibition of lysosomal function with agents such as bafilomycin A1 or 4-amino-quinolines such as chloroquine results in a higher LC3-II:LC3-I ratio, because LC3-II accumulates at the inner autophagosome membrane. However, blockade of autophagy using a dominant-negative ATG4B^C74A^ expression vector, which inhibits autophagy by blocking LC3-I proteolysis [99], or utilizing a dominant-negative form of ULK1 (M92A), which inhibits autophagy at the earliest steps, results in a lower LC3-II:LC3-I ratio, due to decreased LC3-I to LC3-II conversion [77]. An alternative marker of autophagy is the abundance of p62^SQSTM1^, an autophagy cargo receptor and adaptor molecule on the autophagic membrane, which recognizes ubiquitylated proteins targeted for autophagic degradation (Figure 2). Through interaction with LC3-II, p62^SQSTM1^ is continuously degraded by the process of autophagy such that increased or decreased p62^SQSTM1^ abundance (measured by immunoblotting) can serve as a reliable readout of decreased or increased autophagy, respectively [100].

Numerous studies, especially those conducted in genetically engineered mouse (GEM) models, suggest that autophagy serves as a pro-tumorigenic process that contributes to the conversion of normal cells into cancer cells [101,102,103,104]. For example, silencing of ATG5 or ATG7 in GEM models of KRAS^G12D^- or BRAF^V600E^-driven cancer significantly delayed the onset and altered the histopathology of lung or pancreatic cancer [79,80,81]. Moreover, in Braf^V600E^-driven and *Pten* heterozygous melanomas, Atg7 deletion dramatically suppressed tumor formation [104]. However, other studies have suggested that the role of autophagy as a tumor-promoting or tumor-suppressive process may be context-dependent, depending on cooperating genetic alterations, tumor stage/grade, extracellular conditions, and in response to different therapeutic interventions (Figure 2) [82,105]. For example, it is reported that either genetic inhibition of autophagy (ATG5^Null^ or ATG7^Null^) or pharmacological inhibition of lysosome function accelerates the lethal manifestations of KRAS^G12D^/TP53^Null^-driven pancreatic cancers [79].

In the context of normal cells, autophagy protects from the accumulation of misfolded proteins and damaged organelles in order to maintain cellular homeostasis. Activation of autophagy can induce a unique cell death pathway, also known as autophagic cell death (ACD), acting in a tumor-suppressive manner [106,107]. BRAF^V600E^ overexpression by retroviral infection of normal human melanocytes resulted in oncogene-induced senescence (OIS), but was overcome by ATG5 silencing, suggesting that impaired autophagy may contribute to the conversion of normal melanocytes into malignant melanoma cells, at least in vitro [108].

During early stages of tumorigenesis, autophagy may serve to protect initiated cells from accumulating reactive oxygen species (ROS), as well as tissue or genomic damage. Studies in GEM models of *Braf*-mutated lung cancer indicated that expression of BRAF^V600E^ in conjunction with autophagy inhibition through genetic silencing of Atg7 enhances the development of early-stage lung tumors, suggesting a tumor-suppressive role of autophagy at least at early stages in lung tumorigenesis [83]. Furthermore, mice heterozygous for a *Beclin-1^null^* allele, or mice with constitutional *Atg5* or *Atg7* deficiency, all of which are important mediators of autophagy (Figure 2), develop spontaneous tumors [82,109,110]. In early stages of melanomagenesis, autophagy is reported to be decreased compared to either normal melanocytes or benign melanocytic nevi based on reduced expression of ATG5, LC3-B, and Beclin1, and increased accumulation of p62^SQSTM1^, based on immunohistochemical analysis of FFPE specimens [108]. These data suggest that autophagy in non-transformed cells may play a tumor-suppressive role to protect from tumorigenesis by removing damaged cell components, while during early stages of melanoma formation, autophagy inhibition through AGT5 silencing promotes melanoma progression, perhaps through bypassing of oncogene-induced senescence [108,111]. However, based on the level of Beclin1 and LC3 expression, a separate study concluded that the level of autophagy is comparatively low in melanoma in situ [112]. Moreover, p62^SQSTM1^ expression was reported to be elevated in early-stage melanoma compared to benign nevi, suggesting that p62^SQSTM1^ may accumulate in benign nevi due to low autophagic activity [113]. However, due to challenges in extrapolating the level of flux through the autophagy pathway based on static measures of protein expression (e.g., immunohistochemistry or immunoblotting), caution must be exercised in the interpretation of such data.

In the context of BRAF-driven tumorigenesis, either overexpression of normal BRAF (BRAF^WT^) or the mutationally activated oncoprotein kinase BRAF^V600E^ has been reported to increase autophagy in melanoma or colorectal cancer cells, as assessed by changes in the LC3-I:LC3-II ratio, with BRAF^V600E^ having the more pronounced effects on autophagy [114,115,116]. In a similar manner to BRAF, mutationally activated RAS is reported to induce autophagy in a number of cancers [60,101,117], although the presence of a RAS mutation may not be predictive of sensitivity to chloroquine or other autophagy/lysosomotropic agents as it may be context-dependent [118,119]. More generally, sustained activity of the MAPK signaling pathway has been shown to be sufficient for autolysosomal vacuolation [120].

In established tumors, the autophagy machinery is often subverted to fulfill the increased demand for the anabolic metabolism that is required for ongoing cell proliferation, and thereby tumor maintenance [105]. For example, in advanced or metastatic melanoma, Beclin1 and LC3 are reported to be over-expressed compared to early primary lesions, and also correlated with markers of cell proliferation such as Ki67 expression [111,112,121,122]. Furthermore, elevated levels of autophagy as determined based on high LC3 expression in tumor tissue arrays of nearly 1400 tumors from 20 types of cancer also correlated with worse outcome [121]. Furthermore, RAS- or BRAF-driven pancreatic or lung adenocarcinomas are reported to have elevated levels of autophagic flux and are reported to be dependent on autophagy [60,101,102,117]. Autophagy has also been shown to be pro-tumorigenic and essential for oncogenic RAS- or BRAF-induced malignant cell transformation [123,124]. Consistent with these observations, silencing of ATG5 or ATG7 diminishes BRAF^V600E^-driven melanoma or lung cancer in GEM models [83,104,124]. Mechanistically, pharmacological inhibition of autophagy or knockdown of key autophagy genes, such as Beclin1 or Atg7, has been shown to induce cellular senescence or apoptosis, potentially through effects on endosomal maturation [120,121]. Altogether, these observations support the notion that autophagy can have both tumor-promoting or tumor-suppressive effects, and that such effects are likely to be displayed in a context-dependent manner.

## 4. Role of Autophagy in Response to RAS Pathway-Targeted Therapy

Vertical inhibition of BRAF^V600E^→MEK→ERK MAP kinase signaling with combinations of inhibitors of BRAF^V600E^ (vemu-, dab-, or encorafenib) plus MEK1/2 (cobi-, tram-, or binimetinib) are FDA-approved first-line treatment strategies for patients with advanced BRAF^V600^-driven melanoma, non-small-cell lung carcinoma (NSCLC), or thyroid cancer. Interestingly, inhibition of BRAF^V600E^ signaling has been reported to induce autophagy in cancer cell lines [116,125], which raises the question as to how autophagy is altered upon pathway-targeted inhibition of BRAF^V600E^ in cancer cells.

Inhibition of BRAF^V600E^→MEK1/2→ERK1/2 signaling has been shown to induce autophagy in BRAF^V600^-driven melanoma cell lines (Figure 3) [125,126,127]. Furthermore, it was reported that the autophagy regulators LC3 and Beclin1 are expressed at higher levels in *BRAF*-mutated colorectal cancer cell lines, and that pharmacological inhibition of BRAF^V600E^ signaling resulted in reduced expression of these proteins [116]. In *BRAF*-mutated thyroid cancer cells, inhibition of BRAF^V600E^ signaling was reported to induce autophagy as an adaptive response to endoplasmic reticulum (ER) stress, as assessed using immunoblotting analysis in combination with electron microscopy [128]. Hence, although it is likely that *BRAF*-mutated cancer cells have a high baseline level of autophagic flux, it can be further elevated in response to pharmacological inhibition of BRAF^V600E^ signaling. Although not all cancer cells are entirely dependent on autophagy for survival, cancer cells may be more sensitive to autophagy inhibition than normal tissues, which may provide a therapeutic index for cancer therapy [81,129].

## 5. Mechanism of BRAFi-Induced Autophagy

A key question relates to the mechanism by which signaling by the BRAF oncoprotein kinase regulates autophagy. To date, several mechanisms have been proposed, including: (1) direct activation of the MAPK→AMPK/mTOR→ULK1/ATG1 signaling pathway, (2) activation of the endoplasmic reticulum (ER) stress gatekeeper GRP78, and (3) activation of transcriptional factors, such as TFEB and TFE3, two known master regulators of the expression of key autophagy genes, the tumor suppressor p53, or the bromodomain-containing protein 4 (BRD4) (Figure 3) [125,126,127].

Autophagy induction can be orchestrated in response to inhibition of BRAF oncoprotein signaling through the energy stress sensor AMP-activated protein kinase (AMPK) or the nutrient sensor mTORC1. AMPK directly phosphorylates mTORC1 as well as ULK1 complexes [95,130]. Acting through the MEK1/2→ERK1/2 MAPK pathway, both BRAF^V600E^ and KRAS oncoproteins have been shown to inhibit the activity of the protein kinase LKB1, which is a known tumor suppressor [126,131]. ERK1/2-mediated inhibition of LKB1 protein kinase activity leads to reduced signaling through the LKB1→AMPK→ULK1 pathway, allowing for cell proliferation, growth, and survival [77,126,131]. Consequently, pharmacological inhibition of MAPK signaling results in increased flux through the LKB1→AMPK→ULK1 pathway, leading to increased autophagy and inhibition of cell proliferation [77,126,131]. Moreover, AMPK can directly and indirectly inhibit mTORC1 through phosphorylation of the mTORC1 subunit Raptor at Ser792, as well as through phosphorylation of TSC2 at Ser1387, which inhibits the activation of mTORC1 by RHEB-GTP, resulting in autophagy induction [132,133]. Furthermore, ERK1/2 can activate mTORC1 by phosphorylation of Raptor at Ser8, Ser696, and Ser863 [134]. mTORC1 in turn directly phosphorylates and represses the serine/threonine protein kinase ULK1/ATG1 at Ser757 [95]. Consequently, pharmacological inhibition of MAPK or mTORC1 signaling reverses the repressive phosphorylation of ULK1, leading to increased autophagy (Figure 3).

The endoplasmic reticulum (ER) stress response is triggered in response to the accumulation of misfolded proteins, such that the cell can repair the damage or initiate apoptosis [135,136]. The ER stress response is part of the integrated stress response (ISR), which is an evolutionarily conserved intracellular signaling network that helps the cell, tissue, and organism to adapt to a variable environment and maintain health [137]. Mechanistically, autophagy has been shown to be triggered via the ER stress response in BRAF^V600E^ melanoma and thyroid cancer cells [125,128]. In the presence of a BRAF inhibitor, BRAF^V600E^ is reported to bind to the ER stress gatekeeper and chaperone GRP78, resulting in dissociation of GRP78 from the protein kinase RNA-like endoplasmic reticulum kinase (PERK) receptor, which leads to ER expansion, expression of the pro-apoptotic protein CHOP, and phosphorylation of the eukaryotic initiation factor 2 (eIF2α) (Figure 3) [125,138]. eIF2α phosphorylation promotes the translation of the transcription factor ATF4, which regulates the transcription of the essential autophagy genes ATG5 and ATG7 [139], ultimately leading to stress-induced autophagy [125,128]. In BRAF^V600E^ melanoma cells, ER stress-induced autophagy by fenretinide or bortezomib, inhibitors of mTORC signaling or of the proteasome respectively, was elicited to a significantly lesser extent than in non-*BRAF*-mutated melanoma [114,140].

Autophagy can also be activated in response to BRAFi at the transcriptional level, for example through activation of the transcription factor EB (TFEB), the tumor suppressor p53, or the bromodomain-containing protein 4 (BRD4) (Figure 3). It has been reported that oncogenic BRAF^V600E^ signaling results in decreased mTORC1 signaling, leading to increased autophagy [115], speculatively through the AMPK→ULK1 pathway, or through the ERK1/2→TFEB/TFE3 axis, a transcription factor program that can induce autophagy regulating genes. mTORC1 has been shown to phosphorylate TFEB and TFE3, at S142/S211 or at Ser321 respectively, which inhibits their nuclear translocation and sequesters them into the cytoplasm [141,142]. Moreover, it has been reported that BRAF^V600E^ signaling leads to ERK1/2-mediated inactivating phosphorylation of TFEB [127]. Consequently, pharmacological inhibition of BRAF^V600E^ or mTORC1 signaling reverses the repressive phosphorylation of TFEB and TFE3, allowing for their nuclear translocation and initiation transcription of ATG and lysosome biogenesis genes, which ultimately leads to the induction of autophagy [127,143,144,145]. Finally, expression of the DNA stress-induced transcription factor, p53, has been shown to promote autophagy by activating the transcription of autophagy modulating genes, such as several ATG proteins, ULK1 and 2, and the damage-regulated autophagy modulator (DRAM) [146,147,148]. Lastly, the bromodomain-containing protein 4 (BRD4), a BET family protein that binds to chromatin and recruits transcriptional regulators, has been shown to repress the transcription of several crucial autophagy genes (Figure 3) [149].

## 6. Co-Inhibition of Autophagy and Oncogenic BRAF Signaling as a Therapeutic Strategy

Given the dependency of cancer cells on autophagy, either at baseline or in response to anti-cancer therapeutics, it has become an interesting target for pharmacological or genetic inhibition in preclinical models and, more recently, in clinical trials [77,83,101,121,124,150,151,152,153,154,155,156]. However, as a monotherapy, or in combination with standard chemotherapy, pharmacological inhibition of autophagy using the lysosomotropic 4-aminoquinolones chloroquine or hydroxychloroquine has shown incomplete tumor eradication in mouse models, as well as in clinical trials [77,155,157,158].

Since autophagy is increased following inhibition of oncoprotein signaling in several cancer types, recent efforts have focused on combined inhibition of autophagy and cancer type-specific signaling pathways to elicit anti-tumor effects [77,153,154,159,160]. For instance, in BRAF^V600E^-driven lung cancer, co-inhibition of autophagy and MEK1/2 significantly reduced the tumor burden in xenografted mice [77]. In another study, it was shown that silencing of ATG7 suppresses the growth of BRAF^V600E^/PTEN^Null^ melanomas, leading to extended survival of tumor-bearing mice [104]. These data have resulted in a phase I/II clinical trial in which dabrafenib (BRAFi), trametinib (MEKi), and hydroxychloroquine (autophagy/lysosome inhibitor) are being tested in patients with advanced BRAF-mutant melanoma (BAMM: NCT02257424). The phase II portion of the study is currently still ongoing, but the latest results presented at the ASCO meeting in 2018 were promising, with clinical responses noted for a number of patients.

Inhibition of autophagy has also been assessed in combination with inhibition of PI3K→AKT→mTOR signaling in *BRAF*-driven cancers. Pharmacological inhibition of PI3K→AKT→mTOR signaling activated autophagy in both BRAF^V600E^-driven melanoma and colorectal cancer cell lines [116,122]. Combined inhibition of AKT plus autophagy resulted in reduced metabolic activity of metastatic melanoma cells in culture [122,161]. Moreover, it was reported that BRAF^V600E^-driven melanoma cells were less sensitive to mTORC1 inhibitor-mediated autophagy induction (e.g., via rapamycin), suggesting a greater therapeutic benefit for combined inhibition of PI3K→AKT→mTOR signaling plus autophagy in non-BRAF-mutated melanoma cells [114].

Other ways to improve the efficacy of autophagy inhibition for cancer therapy might include strategies to modulate the cancer patient’s dietary intake of essential nutrients. For example, the combination of a leucine-free diet with the autophagy inhibitor chloroquine was reported to reduce the growth of xenografted human melanoma cell lines and triggered caspase-induced apoptosis [162]. Furthermore, autophagy promoted tumor growth by sustaining the levels of the circulating amino acid arginine, and dietary arginine supplementation rescued tumor growth in autophagy-deficient hosts [163]. These studies provide a rationale to investigate dietary restrictions of essential nutrients in autophagy-dependent tumors in certain cancer patients.

## 7. Autophagy Inhibition as a Mechanism to Re-Sensitize BRAF Inhibitor-Resistant Tumors

Although FDA-approved inhibitors of BRAF^V600E/K^ have revolutionized the treatment of certain BRAF-mutated cancers, the twin phenomenon of primary or acquired chemoresistance remains a major clinical challenge. To that end, regulation of autophagy has been implicated as a potential mechanism of drug resistance in BRAF-driven cancer cell lines [125,127,164,165,166]. Paired biopsy samples from BRAF mutant melanoma patients treated with BRAF inhibitors (BRAFi) showed increased levels of autophagy in BRAFi-resistant compared to BRAFi-responsive tumors [125]. Moreover, patients whose melanomas displayed higher levels of BRAFi-induced autophagy by IHC analysis of LC3 expression experienced fewer partial responses (based on >30% shrinkage of tumor) to vemurafenib and shorter progression-free survival [125], suggesting that autophagy is correlated with the ability of cancer cells to adapt to pathway-targeted inhibition of BRAF^V600E^ signaling. Moreover, a correlation was noted between the induction of cytoprotective autophagy and BRAFi-resistance in cultured melanoma cell lines [125]. The same group later showed that BRAFi + MEK1/2i-induced dephosphorylation of ERK1/2 induces the translocation of the MAPK pathway proteins NRAS, BRAF, MEK, and ERK into the endoplasmic reticulum (ER) [166]. This ER translocation is orchestrated by the endoplasmic reticulum (ER) stress gatekeeper GRP78, the scaffolding protein KSR2, and the ER translocase SEC61 [166]. After ER translocation, ERK is re-phosphorylated by the protein kinase RNA-like endoplasmic reticulum kinase (PERK) and translocated into the nucleus, where it phosphorylates and stabilizes the transcription factor ATF4, a transcriptional regulator of the essential autophagy genes ATG5 and ATG7, inducing cytoprotective autophagy [139,166]. Consequently, inhibition of autophagy has been explored as a method to overcome BRAFi resistance. Pharmacologic inhibition of BRAF^V600E^ signaling induced autophagy in BRAFi-resistant melanoma cell lines significantly more than in BRAF-sensitive cell lines [125]. Finally, combined inhibition of autophagy and oncogenic BRAF induced cell death, which led to regression of established xenografted melanomas that were previously judged resistant to BRAFi alone [125].

Similarly, in other BRAF-driven malignancies that have developed resistance to BRAFi, the strategy of sensitizing tumors to BRAFi using autophagy inhibition has been tested. For example, a patient with a BRAF^V600E^-driven ganglioglioma was treated with the combination of vemurafenib plus vinblastine, to which the tumor responded for ~11 months, after which signs of progression were noted. At that time, the vinblastine was discontinued, and the patient was then treated with the combination of vemurafenib plus chloroquine, at which point the patient’s tumor showed clear signs of response that required continued treatment with both vemurafenib plus chloroquine [152]. The same investigators subsequently showed that genetic or pharmacologic inhibition of autophagy inhibition can overcome multiple mechanisms of resistance to BRAFi in *BRAF*-mutated brain tumors [164].

In BRAFi-resistant CRC cell lines, autophagy inhibition using 3-methyladenine (3-MA), which inhibits formation of the autophagophore, sensitized CRC cells to the BRAF^V600E^ inhibitor vemurafenib by triggering apoptotic cell death [116]. Similarly, co-inhibition of autophagy plus MEK1/2 triggered apoptotic cell death in a BRAF-driven colorectal cancer cell line [167]. Moreover, combined inhibition of autophagy plus MEK1/2 promoted regression of a BRAF^V600E^-driven colorectal cancer PDX, resulting in significantly reduced tumor burden compared to the single agents [77]. In *BRAF*-mutated thyroid cancer, where single-agent inhibition of BRAF^V600E^ signaling displays only modest anti-tumor effects [168], it was shown that autophagy was induced upon BRAFi as an adaptive response through endoplasmic reticulum (ER) stress in a MAPK signaling pathway-independent manner [128]. Targeting autophagy in this model sensitized the *BRAF*-mutant thyroid cancer cells to vemurafenib in cell culture and xenografts [128].

## 8. Acquisition of Cancer Mutations Complicate the Picture of How Autophagy Is Regulated

As discussed here, autophagy regulation is biologically complex and can also be context-dependent. Moreover, conclusions regarding the regulation of autophagy by cancer-driving oncoproteins may be confounded by the accumulation of additional genetic alterations over time that promote cancer progression. For example, although mutational activation of *BRAF* is an early event in melanomagenesis, mutations in cooperating oncogenes or tumor suppressors such as PTEN, RAC1, or CDKN2A that often co-occur with *BRAF* mutation may influence the regulation of autophagic flux in malignant melanoma cells [169,170].

The PI3′-lipid phosphatase PTEN is an important melanoma suppressor that acts by antagonizing the accumulation of PI3′-lipids that are important signaling molecules within the cell. PI3′-lipids regulate the activity of the AKT family of protein kinases that are upstream regulators of mTORC activity, a key regulator of autophagy. Based on this model, expression of PTEN would be predicted to increase autophagic flux and also lysosomal mass, as has been shown in human glioma and colon cancer cells [171,172]. Furthermore, PTEN silencing in cancer cells has been reported to decrease autophagy, possibly by enhancing signaling from the insulin or IGF-1 receptor to the PI3′-kinase pathway [173]. Furthermore, it was reported that autophagy is essential for the development of prostate cancer in a mouse model with inducible prostate-specific deficiency in the *Pten* tumor suppressor and autophagy-related-7 (Atg7) genes [174]. Consistent with these observations, inhibition of AKT promoted autophagy and sensitized PTEN^Null^ prostate cancer xenograft tumors to lysosomotropic agents [175].

Mutational alterations in tumor suppressors or proto-oncogenes can inhibit autophagy through effects on the LKB1 > AMPK > mTORC > ULK1 pathway, but are also noted to be able to increase autophagy through effects on the RAF > MEK > ERK MAP kinase pathway. Hence, the combined effects of proto-oncogene activation and tumor suppressor silencing on the level of autophagic flux during cancer initiation and/or progression is likely to be dependent on the cell of origin and the genetics/epigenetics of the tumor and also the local tumor microenvironment [118]. A compelling example involves the tumor suppressor *TP53*, which has been shown to be regulated by both nutrient availability and also by metabolic activity [176]. Although autophagy is reported to promote tumorigenesis by repressing TP53 expression/activity, TP53 is reported to increase transcription of autophagy-related genes [177]. The exact mechanism by which autophagy represses p53 function is unknown. However, it is known that autophagy suppresses oxidative stress response pathways, e.g., by elimination of reactive oxygen species (ROS) that would otherwise activate p53 [178]. In addition, autophagy may prevent DNA damage and p53 activity by providing substrates for DNA replication and repair, as well as maintain the supply of metabolites to prevent AMPK activity, a known activator of p53 [179,180]. In GEM models of KRAS^G12D^-driven pancreatic cancer, either genetic (silencing of ATG5 or ATG7) or pharmacological (hydroxychloroquine) inhibition of autophagy appears to block the progression of low-grade, pre-malignant pancreatic intraepithelial neoplasias (PanINs) to high-grade pancreatic ductal adenocarcinoma (PDA) [79]. However, if TP53 was silenced concomitant with initiation of KRAS^G12D^ expression, inhibition of autophagy enhanced glucose uptake and enrichment of anabolic pathways, promoted PDA progression, and decreased mouse survival [79]. The same group went on to demonstrate that TP53 induces transcription of the *DRAM*, a gene encoding a lysosomal protein that acts as a damage-regulated inducer of autophagy that is critical for TP53-mediated apoptosis [146]. This observation might help to explain why TP53 silencing results in reduced autophagy, and also why inhibition of autophagy might promote progression of KRAS^G12D^-driven PDA in GEM models [79].

In contrast, a separate study reported that PALB2-associated mammary tumorigenesis was delayed upon autophagy impairment through monoallelic loss of *Becn1*, in the context of normal TP53 but not under conditions of conditional TP53 silencing, suggesting that loss of TP53 expression may override or compensate for the reduced fitness in autophagy-impaired tumor cells [181]. Indeed, in GEM models of BRAF^V600E^-driven lung cancer, Atg7 deletion extended the lifespan of mice independently of the expression of TP53 [83]. The same authors further reported that during early tumorigenesis, autophagy had a tumor-suppressive function, but during tumor progression, autophagy can serve to promote tumor progression [83]. In this scenario, it might be speculated that during early stages, TP53 is still intact, and at late stages, TP53 is lost or mutated, resulting in the differential roles of autophagy at different stages of tumorigenesis. Finally, since there are credible reports of TP53 having tumor suppressor functions outside of the nucleus, it remains possible that subcellular location of TP53 expression may influence the regulation of autophagy in manners that are independent of gene transcription [182,183].

Finally, it is highly likely that autophagy occurring in the tumor microenvironment or in the host has an important influence on cancer cell metabolism, with consequences for cell proliferation and survival mechanism, in a manner which may have implications for the measurement and role of autophagy in the behavior of cultured cells. For example, injection of autophagy-proficient mouse melanoma cells into autophagy-deficient mouse hosts with *Atg7* deletion resulted in reduced tumorigenic growth of the cancer cells [163]. In this case, loss of host autophagy was associated with reduced amounts of the conditionally essential amino acid arginine in the circulation, thereby reducing tumorigenic growth of cancer cells lacking expression of the enzyme argininosuccinate synthase 1 (ASS1), which is required for arginine biosynthesis in the cancer cell. Such observations could have implications for the development of predictive biomarkers for the use of pegylated arginase as a cancer therapy. Furthermore, the conditions under which cancer cells are cultured may also have an impact on both the measurement of autophagy and for autophagy dependence in vitro. Most cancer cells were adapted to culture in nutrient-rich media that is supplemented with fetal bovine serum. Hence, the dependency of cancer cells on autophagy in vitro is likely to be different from that of cancer cells in the original tumors from which the cultured cells were derived. Moreover, cancer cell metabolic dependencies in vivo are likely influenced by adaptations through cell-autonomous and non-autonomous mechanisms, such as interactions with cells of the tumor microenvironment, nutrient and oxygen supply, as well as two- vs. three-dimensional growth [184,185]. Moreover, a high mutational burden may influence autophagy and other metabolic pathways in ways that might complicate the picture when it comes to stratification of autophagy-dependent cancers based on cancer gene dependencies. Indeed, cancer cell lines driven by either HRAS or KRAS have been shown to differentially regulate autophagy, indicating that the level of RAS effector pathway activation and signaling “fine-tuning” may have an important influence on the autophagic machinery [116]. Such observations emphasize the importance of using multiple experimental systems for exploring the role of autophagy in cancer maintenance, including the use of conventional 2D or 3D culture systems, cancer-derived organoids, and both genetically engineered or patient-derived xenograft models of cancer.

## 9. Conclusions

Mutational activation of *BRAF* is a driver of numerous cancer types, including both solid and hematologic malignancies [2,3,4,5,6,7]. Both oncogenic signaling by the BRAF oncoprotein kinase and pathway-targeted therapeutics that inhibit BRAF signaling can have complex and context-dependent effects on autophagic flux in various cancer cell types. Moreover, in some cases, combined inhibition of oncogenic signaling plus autophagy has been shown to be an effective strategy for the treatment of cancers driven by oncogenic KRAS or BRAF both in preclinical models and in clinical trials. However, much of this work has relied on the use of 4-amino-quinolones such as chloroquine, that have pleiotropic effects on cancer cell physiology and are not specific and selective inhibitors of autophagy. Both preclinical and clinical data suggest that combined inhibition of autophagy plus specific signaling pathways such as the RAF > MEK > ERK MAP kinase pathway may become a novel and effective treatment strategy for certain cancers driven by BRAF^V600E^, and may potentially be efficacious for a broader group of KRAS-driven cancers. However, since autophagy has been shown to play a tumor-suppressive role in a cancer stage-, tissue-, and/or context-dependent manner, caution must be exercised in the broader clinical deployment of this therapeutic strategy.

## 10. Future Directions

### 10.1. Identification of Biomarkers

The presence of certain *BRAF* mutations (e.g., *BRAF^T1799A^*) in malignancies such as melanoma and lung cancer serves as a predictive biomarker for the clinical deployment of vertical inhibitors of BRAF^V600E^ signaling [186]. However, there is currently no predictive biomarker for the importance of autophagy in either the cancer cell or the tumor microenvironment. It is possible that either pre- or on-treatment plasma/serum samples might reveal such biomarkers that would be useful for patient stratification in clinical trials. Furthermore, whereas inhibition of BRAF^V600E^ signaling can be readily assessed by analysis of pERK1/2 in tumor biopsy specimens or by analysis of *BRAF^T1799A^* circulating tumor DNA in plasma, there remains a need for reliable biomarkers or imaging modalities of autophagy and its inhibition, whether in the tumor and its local microenvironment, or in tumor surrogates such as PBMCs, exosomes, or in serum/plasma, so that the efficacy of autophagy inhibition can be gauged [187].

It is highly likely that some patients will display either primary chemoresistance to combined inhibition of signaling pathways plus autophagy or that they will develop drug-resistant disease after an initial response that will limit the durability of patient responses. Hence, it will be essential to develop both a deeper mechanistic understanding of mechanisms of resistance as well as biomarkers of such situations. This is important since relevant biomarkers of sensitivity/resistance could be used to stratify patients in clinical trials to optimize the likelihood of designing successful clinical trials. This might also allow alternative co-inhibition strategies that might lead to novel combinations of pathway-targeted or conventional cytotoxic chemotherapies plus autophagy inhibitors.

### 10.2. Development of More Specific and Selective Autophagy Inhibitors

Although 4-aminoquinolones such as hydroxychloroquine (HCQ) are relatively inexpensive, FDA-approved drugs that are well-tolerated in most patients, they are pleiotropic in their mechanism(s) of action and are not autophagy-selective agents. Indeed, the mechanism(s) of autophagy inhibition by such lysosomotropic agents have previously thought to be through protonation of accumulating HCQ inside the lysosome, raising the endolysosomal pH and leading to inhibition of lysosomal hydrolases [188]. However, as we currently understand it, chloroquine likely inhibits autophagosomal bulk degradation through blockade of autophagosome-lysosome fusion without affecting the lysosomal acidity [189]. Recent data has indicated that the protein palmitoyl-protein thioesterase 1 (PPT1) may be the target of chloroquine and hydroxychloroquine [190]. Genetic PPT1 depletion resulted in loss of mTORC1-RHEB interaction through reduced interaction of LAMTOR (Ragulator) with the vacuolar-type H^+^-ATPase (v-ATPase), ultimately leading to mTORC1 and autophagy inhibition [190]. The autophagy pathway includes a number of additional, more specific targets such as the ULK1/2 protein kinases, the VPS34 PI3′-kinase, and the ATG4B cysteine protease, to name a few. Hence, once such agents become more readily available, it will be of interest to assess how more specific and selective autophagy inhibitors impact cancer cell physiology and tumor growth, either as single agents or in combination with other anti-cancer agents. [191]. However, it should be noted that the pleiotropic effects of HCQ and similar agents may actually be desirable since they target other processes that may be mission critical in the cancer cell, such as macropinocytosis [192]. It should also be noted that the many proteins in the autophagy machinery also mediate autophagy-independent functions, for example secretion and exocytosis, pathogen inclusion, or immunological memory, as recently reviewed [193], which may prompt unexpected side-effects in patients during autophagy-targeted therapy.

### 10.3. Combining Autophagy Inhibition with Other Anti-Cancer Therapeutics

Although this review has focused on combining autophagy with inhibitors of RAF > MEK > ERK signaling, it is possible that inhibition of other pathways may promote the cancer cells’ dependency on autophagy for survival. One obvious pathway would be the PI3′-kinase pathway, that leads to the accumulation of phosphoinositive-3′-lipids that have pleiotropic effects within the cell. One of the best-characterized effectors are the AKT1-3 family of protein kinases that are known to regulate mTORC activity. Since there are now a number of FDA-approved PI3′-kinase inhibitors in the clinic and at least one pan-AKT inhibitor in late-stage clinical trials, it is possible that such agents may be usefully combined with autophagy inhibitors. Moreover, upstream of both RAF and PI3′-kinase signaling are the large family of receptor tyrosine kinases, such as EGFR, MET, ROS, ALK, and NTRK, that are mutated in a wide range of different cancers and for which there are numerous FDA-approved drugs linked to predictive biomarkers. Finally, one of the Holy Grails of cancer therapy has been the development of direct and effective pharmacological inhibitors of KRAS oncoproteins. The promising clinical activity of covalent inhibitors of KRAS^G12C^ suggests that such agents might soon be tested in combination with autophagy inhibition in relevant preclinical models of lung, pancreas, and colorectal cancers [194,195,196]. Finally, there has been enormous excitement regarding the efficacy of immune checkpoint inhibitors (ICI) such as anti-CTLA4, anti-PD1, or anti-PD-L1. Recent data suggests that inhibition of autophagy can increase the abundance of class I MHC on the surface of pancreas cancer cells [197]. These data suggest strategies of combining ICIs with autophagy inhibitors in clinical trials.

### 10.4. Expanding Clinical Trials to Other BRAF- or RAS-Driven Cancers

If the strategy of combined targeting of BRAF^V600E^ signaling plus autophagy proves to be effective in BRAF-driven melanoma, this could open a therapeutic avenue for other BRAF-driven cancers, such as lung, colorectal, thyroid, and hairy cell leukemia. Moreover, with accumulating data on non-melanoma BRAF-driven cancers, it is expected that the number of clinical trials testing co-targeting of autophagy plus RAF > MEK > ERK signaling will expand in the near future. Furthermore, the preclinical efficacy of the trametinib (MEK1/2i) + hydroxychloroquine combination against KRAS-driven pancreatic ductal adenocarcinoma or NRAS-driven melanoma has already led to multiple clinical trials (NCT03825289, NCT04145297, and NCT03979651) in these areas. Hence, this novel therapy approach may offer a clinical benefit to a broader group of patients with RAS-driven cancers.

## Figures and Tables

**Figure 1 cancers-13-03498-f001:**
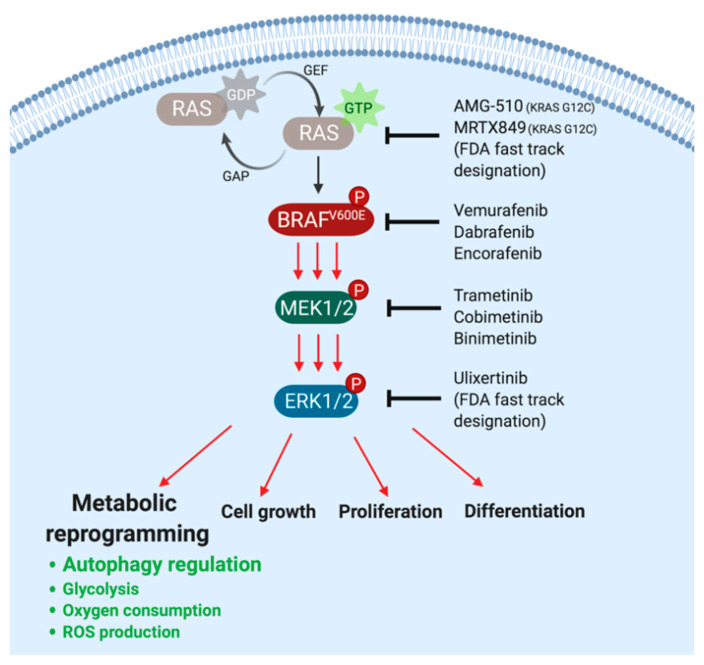
The BRAF→MEK1/2→ERK1/2 signaling pathway and FDA-approved targeted therapy. RAS-GDP to RAS-GTP exchange triggers a BRAF→MEK1/2→ERK1/2 phosphorylation cascade, resulting in metabolic reprograming, cell growth, proliferation, and differentiation. FDA-approved or fast-track-designated inhibitors targeting the MAPK components, as indicated.

**Figure 2 cancers-13-03498-f002:**
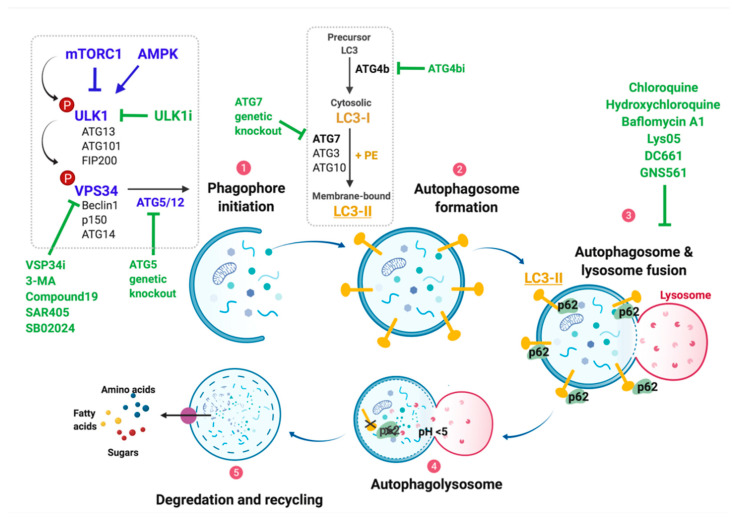
The autophagy machinery and targetable autophagic pathway proteins. Autophagy is a metabolic process of cellular self-digestion in which cellular components, such as abnormal protein aggregates and damaged organelles, serve as recycling substrates in order to generate energy or metabolic precursors, often in response to external stress. The process starts with phagophore initiation, which is regulated by ULK1 and VPS34 complexes, which can be blocked by selective inhibitors, as indicated, as well as ATG activity, resulting in the initiation and elongation of the phagophore membrane. Processing of LC3 by ATG4b as well as several ATGs leads to maturation and finally completion of the autophagosome. The autophagosome then fuses with the lysosome, a step that can be pharmacologically blocked with chloroquine or other compounds, as indicated. In the resulting autophagolysosome, the pH drops to <5, leading to degradation of its collected components.

**Figure 3 cancers-13-03498-f003:**
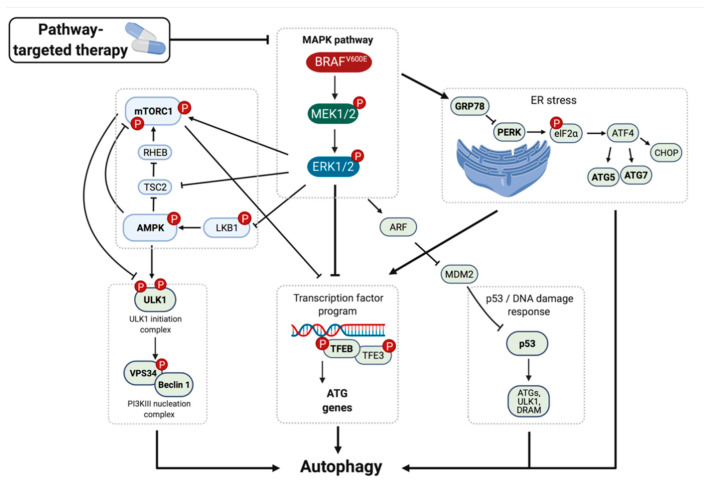
Mechanisms of BRAF inhibitor-induced autophagy. Therapy targeting the MAPK pathway can induce autophagy through several pathways. One way is through direct activation of the MAPK→AMPK/mTOR→ULK1 signaling pathway, leading to activation of VPS34/Beclin1. Another way is through activation of transcription factors, such as TFEB or TFE3, two known master regulators of the expression of key autophagy genes. BRAF inhibition can also induce autophagy through activation of the endoplasmic reticulum (ER) stress gatekeeper GRP78, leading to the activation of ATF4 and transcription of ATG genes. Moreover, the tumor suppressor p53 can induce the transcription of several ATGs, as well as ULK1 and DRAM, resulting in upregulation of autophagy. In theory, MAPK-mediated ARF activation can induce p53, although whether this pathway is used by MAPK-inhibitor exposed cancer cells to upregulate autophagy has not been shown.

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
