# Peer review of "Autophagy Inhibition in BRAF-Driven Cancers"

_cancers, 2021, doi:10.3390/cancers13143498_

Round 1

Reviewer 1 Report

T

The Foth & McMahon review gives a thorough and comprehensive understanding of the BRAF oncogenic driver pathway together with autophagy.

Interestingly the review brings together knowledge from different BRAF driven cancers making the work very sound with interesting prospective in term of therapeutics.

The review refers to important contributions to the field, with about 175 publications, making this review a real over-view of the addressed question.

Three figures illustrate nicely the review. Thought it would have been nice to propose a recap figure showing common and specific mechanisms according to cancer types. This would be an added value to the work.

Author Response

We thank Reviewer 1 for his/her positive comments on our review manuscript submission.  As the reviewer will see from the revised manuscript, we have modified some paragraphs for clarification and improved the flow by rearranging a couple of sections.  With regard to the suggested recap figure, we believe that an attempt to summarize the various different cancer types in their specific or common mechanisms of autophagy induction would not be a trivial endeavor. We are afraid we may not be able to address all those features in one comprehensive figure and are therefore inclined to rely on our prepared Figure 3 in which the most common mechanisms of BRAF inhibitor induced autophagy are summarized. We hope that Reviewer 1 can accept and agree with this arrangement.

Reviewer 2 Report

Comments to the authors

This review from Foth & McMahon is a comprehensive overview of autophagy mechanisms occurring in BRAF-mutant cancers as well as BRAF-inhibitor treated tumors. The authors thereby explain in detail how autophagy can be induced by tumors to promote cell growth and as a resistance mechanism to BRAF-inhibitor therapy. They further elaborate on combining autophagy inhibition and targeted therapy as a treatment regimen for sensitive and BRAF-inhibitor resistant cancers.

Overall, the review is comprehensive and well-described. The figures clearly summarize the data presented in the text. Some paragraphs are a bit too complex for a reader who is not familiar with the research field. Some suggestions for improvement are made below.

Minor points:

1. Pg 2, below Figure 1: Did the authors mean to describe that there are 3 FDA-approved BRAFi/MEKi combinations available for melanoma, referenced by 18-22? If so, could this be added to the sentence?

2. Pg 3, at the end of the first paragraph: Even though the review is focused on autophagy as a resistance mechanism to targeted therapy, could the authors add 1-2 more sentences on other resistance mechanisms?

3. Pg 5, main paragraph: The autophagy mechanism is quite complex. The paragraph would benefit from a couple of additional referrals to Figure 2, where the machinery is nicely summarized.

4. Pg 6: The authors explain the tumor-promoting or tumor-suppressive role of autophagy, but the paragraph does not have a clear line from one to the other: they start with data on the tumor-promoting role, then mention the tumor-suppressive role, then elaborate on the BRAF-mutation as an inducer of autophagy (which is tumor-promoting) and then continue with a tumor-suppressive role. Could this perhaps be rearranged?

5. Pg 6, 3rd paragraph: This sentence is a bit confusing, as it suggests that BRAFV600E expression always leads to oncogene-induced senescence in melanocytes. However, this may only be true for this particular melanoma model.

6. Section 7 would fit better between section 4 and 5.

Author Response

Reviewer: 2

Overall:  This review from Foth & McMahon is a comprehensive overview of autophagy mechanisms occurring in BRAF-mutant cancers as well as BRAF-inhibitor treated tumors. The authors thereby explain in detail how autophagy can be induced by tumors to promote cell growth and as a resistance mechanism to BRAF-inhibitor therapy. They further elaborate on combining autophagy inhibition and targeted therapy as a treatment regimen for sensitive and BRAF-inhibitor resistant cancers. Overall, the review is comprehensive and well-described. The figures clearly summarize the data presented in the text. Some paragraphs are a bit too complex for a reader who is not familiar with the research field. Some suggestions for improvement are made below.

Response:  We thank Reviewer 2 for his/her generally positive overall evaluation of our manuscript. As documented below, we have revised the manuscript to address the reviewer’s concerns. 

Minor points:

Comment 1:Pg 2, below Figure 1: Did the authors mean to describe that there are 3 FDA-approved BRAFi/MEKi combinations available for melanoma, referenced by 18-22? If so, could this be added to the sentence?

Response 1:  We thank the reviewer for the opportunity to clarify this point. We have added “in melanoma” to the sentence.

Comment 2:Pg 3, at the end of the first paragraph: Even though the review is focused on autophagy as a resistance mechanism to targeted therapy, could the authors add 1-2 more sentences on other resistance mechanisms?

Response 2:  We have added a paragraph on resistance mechanisms to BRAF plus MEK1/2 targeted therapy at the end of the first paragraph on page 3, which reads as follows:

“Despite the success of BRAFV600E plus MEK1/2-targeted therapy in melanoma, metastatic NCSLC and thyroid cancer, the durability of patient response is limited either by primary or acquired chemoresistance [32, 33]. Several resistance mechanisms have been described, which often involve re-activation of the MEK1/2®ERK1/2 MAPK pathway, e.g. through changes to the BRAF protein itself, such as dimerization of aberrantly spliced BRAFV600E[9-11, 34], BRAF amplification [35], BRAF fusions [10, 11], BRAF kinase domain duplications [36], or activation of the RAF family member CRAF [37, 38]. Moreover, point mutations in RAS or MEK1 [34, 39-43], as well as upregulation of EGFR, PDGFR or FGFR3 [44, 45], or persistent formation of the eIF4F translation initiation complex [46]have been described to result in re-activation of the MAPK pathway and therapy resistance. Furthermore, PI3K®AKT pathway alterations have been reported to cause resistance to BRAFV600Eplus MEK1/2 inhibition, including point mutations in AKT1 and 3 [47-49], or loss of the tumor suppressor PTEN [50]. Other reported resistance mechanisms include loss of STAG2 and STAG3 [51]or miRNA upregulation[52]. Therapeutic strategies to forestall or overcome drug resistance are currently heavily being investigated, ranging from discontinuous (“drug holiday”) dosing to combinatorial therapy using inhibitors that induce synergistic killing of tumor cells or otherwise prevent re-activation of the MAPK pathway [53, 54].”

Comment 3:Pg 5, main paragraph: The autophagy mechanism is quite complex. The paragraph would benefit from a couple of additional referrals to Figure 2, where the machinery is nicely summarized.

Response 3:  We have added a couple of additional referrals to Figure 2 in the main paragraph describing the autophagic machinery.

Comment 4:Pg 6: The authors explain the tumor-promoting or tumor-suppressive role of autophagy, but the paragraph does not have a clear line from one to the other: they start with data on the tumor-promoting role, then mention the tumor-suppressive role, then elaborate on the BRAF-mutation as an inducer of autophagy (which is tumor-promoting) and then continue with a tumor-suppressive role. Could this perhaps be rearranged?

Response 4: Section 3 (“Autophagy is a recycling mechanism with tumor suppressive or tumor-promoting roles in a context-dependent manner”) has been re-arranged for a better flow, so that the tumor-suppressive role of autophagy is discussed first, followed by the tumor-promoting role.

Comment 5: Pg 6, 3rd paragraph: This sentence is a bit confusing, as it suggests that BRAFV600E expression always leads to oncogene-induced senescence in melanocytes. However, this may only be true for this particular melanoma model.

Response 5: We have edited the sentence in question for clarity purposes. The sentence now reads: “BRAFV600E overexpression by retroviral infection of normal human melanocytes resulted in oncogene-induced senescence (OIS), but was overcome by ATG5 silencing, suggesting that impaired autophagy may contribute to the conversion of normal melanocytes into malignant melanoma cells, at least in vitro [ref].”

Comment 6: Section 7 would fit better between section 4 and 5.

Response 6:  We have moved section 7 (“Mechanism of BRAFi-induced autophagy”) between sections 4 and 5 for a better flow.

Reviewer 3 Report

I commend the authors for a timely and comprehensive review on the topic.

Author Response

Thank you for your comment. We appreciate your time and consideration of our manuscript. We did our best for a timely re-submission of our point-to-point statement.